# The maternal postnatal six-week check in women with epilepsy: Does the prevalence or subsequent postpartum health differ from the general postnatal population?

Kathryn E. Fitzpatrick[1]*, Liza Bowen[2], Yangmei Li[1], Chun Hei Kwok[3,4], Fiona Alderdice[1,5], Suresha Dealmeida[6], Chris Gale[7,8], Sara Kenyon[9], Maria A. Quigley[1], Julia Sanders[10], Dimitrios Siassakos[11,12], Claire Carson[1]

1 NIHR Policy Research Unit in Maternal and Neonatal Health and Care, National Perinatal Epidemiology Unit, Nuffield Department of Population Health, University of Oxford, Oxford, United Kingdom, 2 Population Health Research Institute, St George's, University of London, London, United Kingdom, 3 Big Data Institute, Li Ka Shing Centre for Health Information and Discovery, Old Road Campus, University of Oxford, Oxford, United Kingdom, 4 Applied Health Research Unit, Nuffield Department of Population Health, University of Oxford, Oxford, United Kingdom, 5 School of Nursing and Midwifery, Queen's University Belfast, Belfast, United Kingdom, 6 Bolingbroke Medical Centre, Battersea, London, United Kingdom, 7 Neonatal Medicine, School of Public Health, Faculty of Medicine, Chelsea and Westminster Hospital campus, Imperial College London, London, United Kingdom, 8 Centre for Paediatrics and Child Health, Imperial College London, London, United Kingdom, 9 School of Health Sciences, College of Medical and Dental Sciences, University of Birmingham, Birmingham, United Kingdom, 10 School of Healthcare Sciences, Cardiff University, Cardiff, United Kingdom, 11 Institute for Women's Health, University College London, London, United Kingdom, 12 University College London Hospitals NIHR Biomedical Research Centre, London, United Kingdom

* kate.fitzpatrick@npeu.ox.ac.uk

## Abstract

### Objectives

To examine the prevalence of the maternal postnatal six-week check (SWC) in women with epilepsy compared to a sample of the postnatal population without epilepsy, and assess whether the SWC is associated with health outcomes in the first year postpartum.

### Methods

Clinical Practice Research Datalink Aurum and Hospital Episode Statistics data were used to identify births between January1998-March2020 to women with epilepsy (n=23,533) and a random sample of births to women without epilepsy (n=317,369). The adjusted risk ratio (aRR) for not having a SWC in women with compared to without epilepsy was estimated using modified Poisson regression. The association between receiving a SWC and postpartum health outcomes was assessed using Cox regression.

**Data availability statement:** This study (protocol number 22_002473) was approved through the CPRD Research Data Governance process. All data used in this study were provided by the CPRD. CPRD data governance and the license to use CPRD does not allow the data to be shared. Researchers must seek approval via CPRD's Research Data Governance process to access the data for the purposes of their own research (https://www.cprd.com/data-access). This study is based in part on data from the CPRD obtained under licence from the UK Medicines and Healthcare products Regulatory Agency. The data is provided by patients and collected by the NHS as part of their care and support. Linked data were also provided by the ONS. ONS and HES data © (2021) were re-used with the permission of The Health & Social Care Information Centre. All rights reserved. The interpretation and conclusions contained in this study are those of the authors alone.

**Funding:** This research is funded by the National Institute for Health Research (NIHR) Policy Research Programme, conducted through the NIHR Policy Research Unit in Maternal and Neonatal Health and Care (PR-PRU-1217-21202). The views expressed are those of the authors and not necessarily those of the NIHR or the Department of Health and Social Care. The funders had no role in study design, data collection and analysis, decision to publish, or preparation of the manuscript. FA, CG, SK, MQ, JS, DS, CC received the funding (PR-PRU-1217-21202). SK is part funded by the NIHR Applied Research Collaboration (ARC) West Midlands (NIHR200165).

**Competing interests:** The authors have declared that no competing interests exist.

## Results

The likelihood of not having a SWC did not differ between those with and without epilepsy (42.7% vs 43.4%, aRR = 1.01, 95%CI = 0.99–1.03). Among all women, not having a SWC was associated with a lower subsequent likelihood of being prescribed prophylactic (aHR = 0.59, 95%CI = 0.58–0.60) and emergency (aHR = 0.95, 95%CI = 0.91–0.99) contraception and having urinary and/or faecal incontinence (aHR = 0.67, 95%CI = 0.61–0.73) or dyspareunia, perineal and/or pelvic pain (aHR = 0.70, 95%CI = 0.65–0.75) recorded in the year postpartum, with no evidence these associations differed according to whether a woman had epilepsy. Not having a SWC was also associated with a lower likelihood of having depression and/or anxiety recorded in the first year postpartum among those without (aHR = 0.86, 95%CI = 0.84–0.89) but not with epilepsy (aHR = 1.01, 95%CI = 0.93–1.09). The SWC was not associated with epilepsy relevant outcomes (Accident and emergency visits or unplanned hospital admission for epilepsy, mortality).

## Conclusions

Around 2 in every 5 women had no evidence of a maternal SWC, with no evidence epileptic women had a different prevalence to the general postnatal population. The maternal SWC may play a role in increasing the use of contraception and the detection or treatment of adverse health outcomes in the first year postpartum.

## Introduction

A maternal health check with a general practitioner (GP) at 6–8 weeks postpartum has long been recommended [1], although it only became an essential service in England under the GP contract in February 2020 [2]. UK guidelines [3] recommend that this maternal postnatal 'six-week check' (SWC) should focus on: mental health and general wellbeing; return to physical health and identification of pelvic health issues; family planning and contraception; and pregnancy-related conditions which may need ongoing management. However, there is limited evidence for the effectiveness of the maternal SWC in improving women's longer-term health either in the general postnatal population or in those with pre-existing medical conditions such as epilepsy, who have a higher risk of experiencing adverse outcomes. It is also unknown whether women who have pre-existing medical conditions like epilepsy are more or less likely to have this general postpartum health check.

With a prevalence of 0.5–1%, epilepsy is one of the most common neurological disorders to affect women of childbearing age [4]. Planning pregnancies, and therefore provision of appropriate contraception, in addition to careful management during pregnancy and the postnatal period is particularly important for women with epilepsy because the condition and some anti-epileptic drugs (AEDs) can have serious consequences for women and their babies. Due to the teratogenicity of the AED sodium valproate, the Medicines and Healthcare Products Regulatory Agency

have since May 2018 issued guidance that this medicine should only be used in women of childbearing potential if a pregnancy prevention programme is in place. The Medicines and Healthcare Products Regulatory Agency have also recently issued this guidance for the AED topiramate [5]. UK guidelines [4] also recommend that women taking sodium valproate or other AED polytherapy should have a discussion with an epilepsy specialist on the risks and benefits of continuing or changing the AED prior to conception, with the aim being adequate management of the condition in pregnancy avoiding exposure to sodium valproate and AED polytherapy where possible. While the majority of women with epilepsy will experience an uncomplicated pregnancy, there is some evidence that they are at increased risk of complications during pregnancy and the postnatal period including pre-eclampsia, preterm birth, and postnatal depression and anxiety [6–8]. Furthermore, although rare, sudden unexplained death in epilepsy is an increasing cause of maternal mortality in the UK [9]. UK guidelines [4] recommend that mothers with epilepsy should be well supported in the postnatal period to ensure that triggers of seizure deterioration are minimised and that AEDs should be continued postnatally, although the dose of AED may need to be reviewed. While women with epilepsy should receive multi-specialist care during pregnancy, it is important that they also receive standard postpartum care. A key aspect of postpartum care is the SWC delivered by the GP.

This study aimed to:

1) Examine whether women with a diagnosis of epilepsy prior to giving birth are more or less likely to receive their SWC than a general postnatal population sample of women without epilepsy.

2) Assess whether receiving a SWC is associated with general postnatal health outcomes in the first year postpartum such as being prescribed contraception or having evidence of urinary and/or faecal incontinence, and if this differs according to whether or not a women has epilepsy.

3) Assess, among the women with epilepsy, whether receiving a SWC is associated with epilepsy relevant health outcomes in the first year postpartum including any accident and emergency (A&E) visits or unplanned hospital admission for epilepsy, mortality or recording of a pregnancy prevention plan in the GP records.

## Methods

### Study design and data sources

A population-based cohort study was conducted using the Clinical Practice Research Datalink (CPRD) Aurum database in addition to linked Hospital Episode Statistics (HES), English Index of Multiple Deprivation (IMD) and Office for National Statistics (ONS) death registration data. As of September 2023, CPRD Aurum contained de-identified patient-level primary care records including data on demographics, health-related behaviours, symptoms, tests, diagnoses, referrals and prescriptions for around 20% of the UK population [10]. Included patients are broadly representative of the English population in terms of geographical spread, deprivation, ethnicity, age and gender [11,12]. Within CPRD Aurum, the pregnancy register is an algorithm that uses antenatal, birth and postnatal events recorded in the primary care records to identify pregnancy episodes and their outcomes, with the algorithm described in detail elsewhere [13]. HES contains patient demographic, clinical and administrative data on all hospital admissions, outpatient appointments and A&E attendances at NHS hospitals in England. The IMD is an area-based measure of deprivation. Linkage to HES, IMD and ONS death registration data is available for around 80% of CPRD Aurum patients [10]. A complete description of the data sources, codes, and database fields used is provided in S1 Table.

### Study population

All pregnancies ending in a live birth or stillbirth between 1st January 1998 and 31st March 2020 according to the CPRD pregnancy register or HES admitted patient care, to women aged 11–49 years, with acceptable research quality data,

eligible for linkage to HES, and actively registered at an English CPRD Aurum practice for at least one year before and 12 weeks after pregnancy were identified. From this population, all pregnancies to women with epilepsy were included in the study along with a random 10% sample of pregnancies to women without epilepsy. Women were considered as having epilepsy if they had a diagnosis of epilepsy in their primary care records at any time up to the date they gave birth, using a rigorously developed codelist from a recent study examining the incidence and prevalence of epilepsy in the UK [14].

## Maternal SWC

Based on a previously developed codelist [15], women were considered as having a maternal SWC if they had codes in their primary care records specifically describing maternal SWCs or had codes indicating a possible maternal SWC such as 'postnatal examination observations' between 4–12 weeks postpartum.

## Outcomes

A number of health outcomes in the first year postpartum of particular relevance to women with epilepsy were investigated including whether the woman: had any A&E visits or unplanned hospital admissions for epilepsy; died according to her primary care records or ONS death registration data; and had a recording of a pregnancy prevention plan in her primary care records. Several more general postnatal health outcomes in the first year postpartum were also investigated in women with and without epilepsy, including whether the woman had the following in their primary care records: prescribed female prophylactic contraception (e.g., implant and intrauterine device or contraceptive pill) or emergency contraception (levonorgestrel or ulipristal acetate); had evidence of depression and/or anxiety; evidence of urinary and/or faecal incontinence; and evidence of dyspareunia, perineal pain or pelvic pain. The analysis of any A&E visit for epilepsy was restricted to pregnancies endings between 1st March 2007 and 31st March 2019 due to the availability of linked A&E data. Women who had codes in their primary care records at or after the end of their pregnancy but prior to the start of follow up (see statistical analysis) indicating they could not become pregnant (e.g., because they had a caesarean hysterectomy) were excluded from the analysis of prophylactic and emergency contraception as well as recording of a pregnancy prevention plan. The analysis of recording of a pregnancy prevention plan was additionally restricted to women who were prescribed sodium valproate at or in the three months prior to the start of follow up who gave birth on or after 1st May 2018, the date the Medicines and Healthcare Products Regulatory Agency stated that this medicine should only be used in women of childbearing potential if a pregnancy prevention programme was in place. Codelists to identify the outcomes of interest were created using the following process. All potential terms that may be used to identify the outcome in question were compiled, informed by clinical input and searching published articles and online codelist repositories. This was then used as a basis for searching all the available codes to identify a list of potential codes to use, with further clinical input sought to finalise the codes.

## Other variables

Information on the following maternal socio-demographic characteristics was extracted from the primary care and/or linked data: maternal age at delivery, ethnic group, geographic region of the woman's GP practice, and IMD corresponding to the woman's postcode of residence. Information on the following pregnancy/birth characteristics for the index pregnancy/birth was also extracted from the primary care and/or linked data: parity, mode of birth, whether the women had a multifetal pregnancy, experienced gestational hypertension or pre-eclampsia, and whether she had a preterm birth (<37 weeks gestation). Finally, information on measures of prior health care utilisation/medical history were extracted including the number of GP contacts a woman had in the year before pregnancy, and whether the woman had evidence of any of the following at any point between the year before pregnancy and prior to the start of follow up: prescribed female prophylactic

contraception or emergency contraception; had a depression and/or anxiety diagnosis recorded in their primary care records; had urinary and/or faecal incontinence recorded in their primary care records; had dyspareunia, perineal pain or pelvic pain recorded in their primary care records; and, for women with epilepsy, whether they had any A&E visits or unplanned hospital admissions for epilepsy.

## Statistical analysis

Descriptive statistics were used to compare the characteristics of the study population by whether or not women had epilepsy, and the characteristics of the population with epilepsy and without epilepsy according to whether or not they had a maternal SWC. The overall and annual proportion of pregnancies to women with and without epilepsy who had evidence of a maternal SWC was calculated. As the SWC is a common outcome, modified Poisson regression was used to estimate the risk ratio for not having a SWC in women with compared to without epilepsy. Cox proportional hazards models were then used to assess among all women whether having a SWC affected the subsequent likelihood of having contraception prescribed or general adverse health outcomes detected or treated in the first year postpartum, and whether this differed according to whether or not the woman had epilepsy. The latter was assessed by fitting interaction terms to the fully adjusted models. Cox proportional hazards models were also used to assess, among the women with epilepsy, whether having a SWC affected the subsequent likelihood of having epilepsy relevant health outcomes in the first year postpartum. All models were first adjusted for year of birth to account for temporal changes (model A). To examine the relative influence of maternal socio-demographic, pregnancy/birth characteristics and prior health care utilisation/medical history on the associations in question, models were then adjusted in hierarchical fashion: model B was adjusted for maternal socio-demographic characteristics; model C was additionally adjusted for pregnancy/birth characteristics; and model D was additionally adjusted for measures of prior health care utilisation/medical history. All covariates adjusted for were determined a priori based on pre-existing hypotheses or evidence [16–21] on what factors are thought to potentially confound or explain the associations under investigation.

For those who had a SWC, follow up started on the date of their SWC. For those without a SWC, a random follow up start date, herein referred to as the index date, was assigned based on the distribution of the dates recorded in those who had a SWC. Follow-up ended on the earliest of the following: date the outcome of interest was first recorded during follow-up, date woman could no longer become pregnant if the outcome of interest was contraception or recording of a pregnancy prevention plan, date of death, date one year postpartum or end of data collection. For outcomes extracted from primary care, the end of data collection was the date a woman's registration at the CPRD contributing practice ended or the date of the most recent CPRD data collection for the practice, whichever came first. For outcomes extracted from HES admitted patient care, HES A&E, and ONS death registration the end of data collection was 31st March 2021, 31st March 2020 and 29th March 2021 respectively, the latest dates linked data from these sources was available. The proportional hazards assumption was verified visually using log–log plots. Robust standard errors were used to account for the lack of independence in the data of women who had more than one eligible birth in the study period.

We present a complete case analysis. For the SWC and diagnoses of interest, the absence of the codes in question was taken to mean that there was no evidence that these had occurred. Where data items were missing for socio-demographic or birth characteristics, we describe the proportion missing and compared the characteristics of the complete case and whole study population (see S2 Table). All analyses were conducted in Stata 17MP (Statacorp, College Station, TX, United States).

## Ethics

This study (protocol number 22_002473) was approved through the CPRD Research Data Governance process. All the data used in this study was anonymised. The CPRD has generic ethical approval from the Health Research Authority to support research studies using anonymised patient data without the need for written or oral consent following approval through the CPRD Research Data Governance process.

**Patient and public involvement**

This study was supported by two mothers with epilepsy who have had experience of postnatal care in England, recruited with the help of the Epilepsy Action. We consulted our patient and public involvement group during the design of the study, they have commented on the findings, and have contributed to the dissemination plan.

## Results

### Characteristics of study population with and without epilepsy

A total of 23,533 pregnancies to women with epilepsy and 317,369 pregnancies to women without epilepsy were included in the study (S1 Fig). The women with epilepsy were more likely than those without epilepsy to be of White ethnicity, living in a more deprived area, be nulliparous, and have had pregnancy-induced hypertension or pre-eclampsia, an assisted vaginal, elective or emergency caesarean birth, and given birth preterm (Table 1). They were also more likely to have had a higher number of GP contacts in the year before pregnancy, and have had a depression and/or anxiety diagnosis recorded between the year before pregnancy and prior to the SWC/index date.

### Maternal SWC in population with and without epilepsy

Overall, 57.3% of the pregnancies to women with epilepsy and 56.6% of the pregnancies to women without epilepsy had evidence of a maternal SWC, with the prevalence of the maternal SWC increasing over time in both groups from around 37% in 1998 to around 60% in 2019 (Fig 1). There was no evidence that the likelihood of not having a SWC differed between those with and without epilepsy (Table 2).

### Characteristics of study population with and without epilepsy according to whether or not they had a maternal SWC

Among the study population with and without epilepsy, there were differences in the characteristics of those who did and did not have a maternal SWC (Table 3): women who did not have their SWC were more likely to be younger, living in a more deprived area, be multiparous, and to have had an unassisted vaginal birth, and given birth preterm. They were also more likely to have had a lower number of GP contacts in the year before pregnancy and were slightly more likely to have been prescribed prophylactic contraception between the year before pregnancy and prior to the SWC/index date. They were less likely than the women who had a SWC to be of White ethnicity, and were slightly less likely to have had a depression and/or anxiety diagnosis, urinary and/or faecal incontinence and dyspareunia, perineal and/or pelvic pain recorded between the year before pregnancy and prior to the SWC/index date. Among the population with epilepsy, the women who did not have their SWC were also more likely to have had an unplanned hospital admission for epilepsy between the year before pregnancy and prior to the SWC/index date.

### Maternal SWC and general postnatal health outcomes

Among the whole study population, having adjusted for year of birth, socio-demographic, pregnancy/birth characteristics, and prior healthcare utilisation/medical history (model D), women who did not have a SWC had a significantly lower likelihood than those who did have a SWC of being prescribed prophylactic and emergency contraception at or after the SWC/index date up to 1 year postpartum (Table 4). They also had a significantly lower likelihood of having depression and/or anxiety, urinary and/or faecal incontinence or dyspareunia, perineal and/or pelvic pain recorded at or after the SWC/index date up to 1 year postpartum. There was no evidence these associations differed according to whether or not the woman had epilepsy (S3 Table) with the exception of the association between maternal SWC and depression and/or anxiety: in the fully adjusted model not having compared to having a SWC was associated with a significantly lower likelihood of

**Table 1. Characteristics of study population by whether or not women had epilepsy.**

| | Pregnancies to women without epilepsy n (%)ᵃ unless otherwise stated N=317,369 | Pregnancies to women with epilepsy n (%)ᵃ unless otherwise stated N=23,533 |
|---|---|---|
| *Maternal socio-demographic characteristics* | | |
| **Age at delivery in years** | | |
| <20 | 14,136 (4.5) | 1,099 (4.7) |
| 20-24 | 47,349 (14.9) | 3,788 (16.1) |
| 25-29 | 83,251 (26.2) | 6,278 (26.7) |
| 30-34 | 100,217 (31.6) | 7,025 (29.9) |
| 35-39 | 58,344 (18.4) | 4,192 (17.8) |
| ≥40 | 14,072 (4.4) | 1,151 (4.9) |
| **Median (IQR) age at delivery in years** | 30.7 (26.3,34.6) | 30.4 (26.0,34.6) |
| **Ethnic group** | | |
| White | 258,818 (82.6) | 21,012 (90.0) |
| Asian or Asian British | 31,106 (9.9) | 1,305 (5.6) |
| Black, African, Caribbean or Black British | 14,411 (4.6) | 572 (2.4) |
| Mixed or multiple ethnic groups | 4,177 (1.3) | 294 (1.3) |
| Other ethnic group | 4,985 (1.6) | 164 (0.7) |
| *Missing (%)* | *1.2* | *0.8* |
| **Geographic region** | | |
| North East, Yorkshire and the Humber | 21,011 (6.6) | 1,963 (8.3) |
| North West | 62,008 (19.5) | 4,779 (20.3) |
| Midlands | 61,530 (19.4) | 4,527 (19.2) |
| East of England | 13,762 (4.3) | 938 (4.0) |
| London | 57,483 (18.1) | 3,501 (14.9) |
| South East | 62,824 (19.8) | 4,676 (19.9) |
| South West | 38,751 (12.2) | 3,149 (13.4) |
| **IMD** | | |
| 1 (least deprived) | 58,951 (18.6) | 3,474 (14.8) |
| 2 | 57,513 (18.2) | 4,111 (17.5) |
| 3 | 56,744 (17.9) | 4,068 (17.3) |
| 4 | 64,945 (20.5) | 5,175 (22.0) |
| 5 (most deprived) | 78,094 (24.7) | 6,662 (28.4) |
| *Missing (%)* | *0.4* | *0.2* |
| *Pregnancy/birth characteristics* | | |
| **Parity** | | |
| 0 | 96,548 (30.4) | 7,686 (32.7) |
| ≥1 | 220,821 (69.6) | 15,847 (67.3) |
| **Multifetal pregnancy** | 4,576 (1.4) | 351 (1.5) |
| **Gestational hypertension or pre-eclampsia** | 21,577 (6.8) | 1,946 (8.3) |
| **Mode of birth** | | |
| Emergency caesarean section | 36,954 (12.7) | 3,156 (14.6) |
| Elective caesarean section | 32,160 (11.0) | 2,769 (12.8) |
| Assisted vaginal birth | 31,053 (10.7) | 2,474 (11.4) |
| Unassisted vaginal birth | 190,601 (65.4) | 13,148 (60.8) |
| Other | 601 (0.2) | 62 (0.3) |

*(Continued)*

**Table 1.**  (Continued)

| | Pregnancies to women without epilepsy n (%)ᵃ unless otherwise stated N = 317,369 | Pregnancies to women with epilepsy n (%)ᵃ unless otherwise stated N = 23,533 |
|---|---|---|
| *Missing (%)* | *8.2* | *8.2* |
| **Preterm birth (<37 weeks of gestation)** | 22,212 (7.4) | 1,980 (8.8) |
| *Missing (%)* | *5.2* | *4.4* |
| *Prior health care utilisation/medical history* | | |
| **Number of GP contacts in the year before pregnancy** | | |
| 0 | 51,759 (16.3) | 2,757 (11.7) |
| 1-3 | 94,044 (29.6) | 5,427 (23.1) |
| 4-9 | 114,360 (36.0) | 8,776 (37.3) |
| ≥10 | 57,206 (18.0) | 6,573 (27.9) |
| **Median (IQR) number of GP contacts in the year before pregnancy** | 4.0 (1.0,8.0) | 6.0 (2.0,10.0) |
| **Prescribed prophylactic contraception at any point between the year before pregnancy and prior to the SWC or index date** | 118,365 (37.3) | 8,964 (38.1) |
| **Prescribed emergency contraception at any point between the year before pregnancy and prior to the SWC or index date** | 9,724 (3.1) | 892 (3.8) |
| **Depression &/or anxiety diagnosis at any point between the year before pregnancy and prior to the SWC or index date** | 27,642 (8.7) | 2,991 (12.7) |
| **Urinary &/or faecal incontinence at any point between the year before pregnancy and prior to the SWC or index date** | 2,449 (0.8) | 256 (1.1) |
| **Dyspareunia, perineal &/or pelvic pain at any point between the year before pregnancy and prior to the SWC or index date** | 7,463 (2.4) | 692 (2.9) |

ᵃPercentage of those with complete data.

Abbreviations: GP, General practitioner; IMD, Index of Multiple Deprivation; IQR, interquartile range; SWC, postnatal six-week check.

having depression and/or anxiety at or after the SWC/index date up to 1 year postpartum in those without epilepsy (aHR 0.86 95% CI 0.84–0.89, p < 0.001) but there was no significant association seen in those with epilepsy (aHR 1.01 95% CI 0.93–1.09, p = 0.815).

## Maternal SWC and epilepsy relevant health outcomes

Out of the 436 pregnancies ending on or after 1st May 2018 to women with epilepsy who were prescribed sodium valproate at or in the three months prior to the start of follow up and who had no codes in their primary care records at or after the end of their pregnancy but prior to the start of follow up indicating they could not become pregnant, none had a recording of a pregnancy prevention plan in their primary care records. This outcome was therefore not considered further. Among the population with epilepsy, having only adjusted for year of birth (model A), women who did not have a SWC had a significantly higher likelihood than those who did have a SWC of having an unplanned hospital admission for epilepsy at or after the SWC/index date up to 1 year postpartum (Table 5). However, this association was attenuated and no longer statistically significant having additionally adjusted for socio-demographic factors (model B). Among the population with epilepsy, no significant differences were seen between those who did and did not have a SWC in terms of their likelihood of having an A&E visit for epilepsy or dying at or after the SWC/index date up to 1 year postpartum.

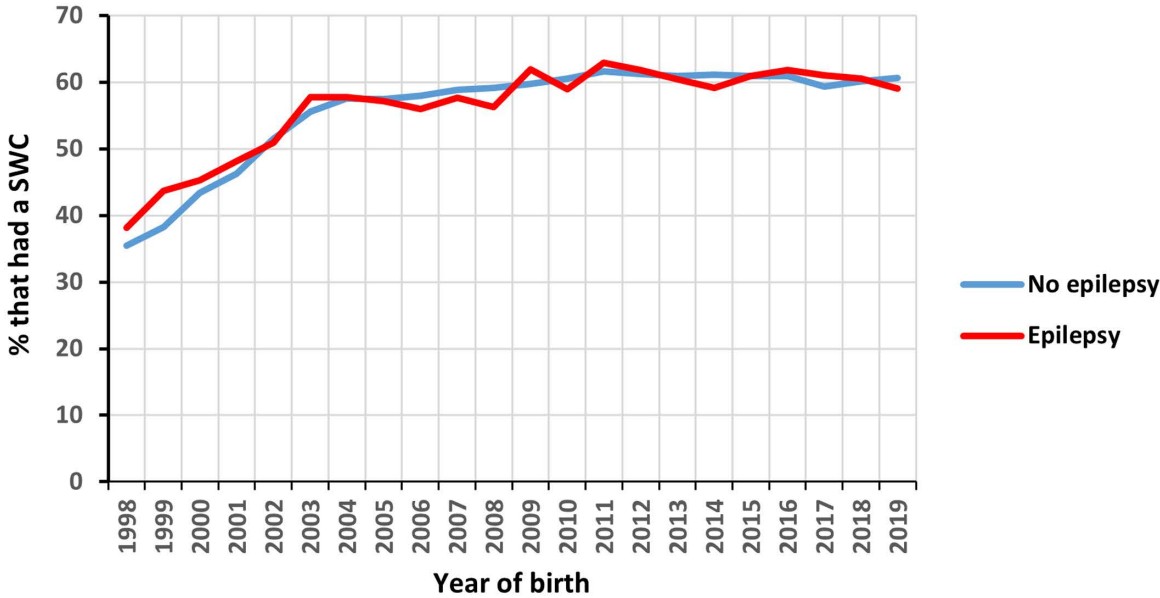

**Fig 1. Percentage of pregnancies to women with and without epilepsy who had evidence of a maternal SWC over time.** Abbreviations: SWC, postnatal six-week check.

**Table 2. Risk ratio of not having a maternal SWC in population with epilepsy compared to without epilepsy.**

|  | Unadjusted RR (95% CI)¥ | Model A* RR (95% CI)¥ | Model B** RR (95% CI)¥ | Model C*** RR (95% CI)¥ | Model D**** RR (95% CI)¥ |
|---|---|---|---|---|---|
| **Pregnancies to women without epilepsy** | 1 | 1 | 1 | 1 | 1 |
| **Pregnancies to women with epilepsy** | 0.99 (0.97-1.01) p = 0.302 | 1.00 (0.98-1.01) p = 0.638 | 0.99 (0.97-1.01) p = 0.223 | 0.99 (0.97-1.01) p = 0.283 | 1.01 (0.99-1.03) p = 0.295 |

*Model A adjusted for year of birth only.

**Model B adjusted for year of birth and maternal socio-demographic characteristics (age at delivery, ethnic group, geographic region & IMD).

***Model C adjusted for variables in Model B and additionally adjusted for pregnancy/birth characteristics (parity, multifetal pregnancy, gestational hypertension or pre-eclampsia, mode of birth & preterm birth).

****Model D adjusted for variables in Model C and additionally adjusted for prior health care utilisation/medical history (number of GP contacts in the year before pregnancy as well as whether had any of the following at any point in between the year before pregnancy and prior to the SWC or index date: prescribed prophylactic contraception, prescribed emergency contraception, had depression &/or anxiety diagnosed, had urinary &/or faecal incontinence recorded, or had dyspareunia, perineal &/or pelvic pain recorded).

¥Analysis restricted to the population who did not have missing data for any of the covariates included in model D (n = 293,272).

Abbreviations: CI, confidence interval; GP, General practitioner; IMD, Index of Multiple Deprivation; RR, Risk ratio; SWC, postnatal six-week check.

## Discussion

### Main findings

This population-based cohort study found that just under 3 in every 5 women giving birth in the period between 1998 and early 2020 had evidence of a maternal SWC. While women with epilepsy were as likely as the general postnatal population sample of women without epilepsy to have evidence of this check, there appears to be other disparities in the provision or uptake. For example, women who were younger, living in a more deprived area, and who gave birth preterm were more likely to not have a SWC. Among all women, not having a SWC was associated with a lower subsequent likelihood

**Table 3. Characteristics of population with epilepsy and without epilepsy by whether or not they had a maternal SWC.**

| | Pregnancies to women with epilepsy | | Pregnancies to women without epilepsy | |
|---|---|---|---|---|
| | Had a SWC n (%)[a] unless otherwise stated N = 13,481 | Did not have a SWC n (%)[a] unless otherwise stated N = 10,052 | Had a SWC n (%) [a] unless otherwise stated N = 179,713 | Did not have a SWC n (%) [a] unless otherwise stated N = 137,656 |
| *Maternal socio-demographic characteristics* | | | | |
| **Age at delivery in years** | | | | |
| <20 | 538 (4.0) | 561 (5.6) | 6,642 (3.7) | 7,494 (5.4) |
| 20-24 | 2,076 (15.4) | 1,712 (17.0) | 24,513 (13.6) | 22,836 (16.6) |
| 25-29 | 3,533 (26.2) | 2,745 (27.3) | 46,668 (26.0) | 36,583 (26.6) |
| 30-34 | 4,217 (31.3) | 2,808 (27.9) | 59,294 (33.0) | 40,923 (29.7) |
| 35-39 | 2,526 (18.7) | 1,666 (16.6) | 34,850 (19.4) | 23,494 (17.1) |
| ≥40 | 591 (4.4) | 560 (5.6) | 7,746 (4.3) | 6,326 (4.6) |
| **Median (IQR) age at delivery in years** | 30.7 (26.3,34.6) | 30.0 (25.5,34.5) | 31.0 (26.8,34.8) | 30.2 (25.7,34.4) |
| **Ethnic group** | | | | |
| White | 12,147 (90.7) | 8,865 (89.1) | 149,167 (83.9) | 109,651 (80.7) |
| Asian or Asian British | 710 (5.3) | 595 (6.0) | 16,065 (9.0) | 15,041 (11.1) |
| Black, African, Caribbean or Black British | 273 (2.0) | 299 (3.0) | 7,599 (4.3) | 6,812 (5.0) |
| Mixed or multiple ethnic groups | 171 (1.3) | 123 (1.2) | 2,276 (1.3) | 1,901 (1.4) |
| Other ethnic group | 95 (0.7) | 69 (0.7) | 2,599 (1.5) | 2,386 (1.8) |
| *Missing (%)* | *0.6* | *1.0* | *1.1* | *1.4* |
| **Geographic region** | | | | |
| North East, Yorkshire and the Humber | 1,062 (7.9) | 901 (9.0) | 11,072 (6.2) | 9,939 (7.2) |
| North West | 2,576 (19.1) | 2,203 (21.9) | 32,905 (18.3) | 29,103 (21.1) |
| Midlands | 2,542 (18.9) | 1,985 (19.7) | 32,562 (18.1) | 28,968 (21.0) |
| East of England | 599 (4.4) | 339 (3.4) | 8,801 (4.9) | 4,961 (3.6) |
| London | 1,989 (14.8) | 1,512 (15.0) | 33,275 (18.5) | 24,208 (17.6) |
| South East | 2,965 (22.0) | 1,711 (17.0) | 38,957 (21.7) | 23,867 (17.3) |
| South West | 1,748 (13.0) | 1,401 (13.9) | 22,141 (12.3) | 16,610 (12.1) |
| **IMD** | | | | |
| 1 (least deprived) | 2,213 (16.4) | 1,261 (12.6) | 37,218 (20.7) | 21,733 (15.9) |
| 2 | 2,544 (18.9) | 1,567 (15.6) | 35,177 (19.6) | 22,336 (16.3) |
| 3 | 2,374 (17.6) | 1,694 (16.9) | 33,504 (18.7) | 23,240 (17.0) |
| 4 | 2,977 (22.1) | 2,198 (21.9) | 35,600 (19.8) | 29,345 (21.5) |
| 5 (most deprived) | 3,357 (24.9) | 3,305 (33.0) | 38,038 (21.2) | 40,056 (29.3) |
| *Missing (%)* | *0.1* | *0.3* | *0.1* | *0.7* |
| *Pregnancy/birth characteristics* | | | | |
| **Parity** | | | | |
| 0 | 4,526 (33.6) | 3,160 (31.4) | 55,440 (30.8) | 41,108 (29.9) |
| ≥1 | 8,955 (66.4) | 6,892 (68.6) | 124,273 (69.2) | 96,548 (70.1) |
| **Multifetal pregnancy** | 187 (1.4) | 164 (1.6) | 2,549 (1.4) | 2,027 (1.5) |
| **Gestational hypertension or pre-eclampsia** | 1,168 (8.7) | 778 (7.7) | 12,679 (7.1) | 8,898 (6.5) |
| **Mode of birth** | | | | |
| Emergency caesarean section | 1,922 (14.9) | 1,234 (14.1) | 21,687 (12.7) | 15,267 (12.6) |
| Elective caesarean section | 1,662 (12.9) | 1,107 (12.7) | 19,271 (11.3) | 12,889 (10.7) |
| Assisted vaginal birth | 1,542 (12.0) | 932 (10.7) | 19,051 (11.2) | 12,002 (9.9) |
| Unassisted vaginal birth | 7,721 (60.0) | 5,427 (62.1) | 110,368 (64.7) | 80,233 (66.5) |

*(Continued)*

**Table 3.** (Continued)

| | Pregnancies to women with epilepsy | | Pregnancies to women without epilepsy | |
|---|---|---|---|---|
| | Had a SWC n (%)ᵃ unless otherwise stated N=13,481 | Did not have a SWC n (%)ᵃ unless otherwise stated N=10,052 | Had a SWC n (%) ᵃ unless otherwise stated N=179,713 | Did not have a SWC n (%) ᵃ unless otherwise stated N=137,656 |
| Other | 27 (0.2) | 35 (0.4) | 256 (0.2) | 345 (0.3) |
| *Missing (%)* | *4.5* | *13.1* | *5.1* | *12.3* |
| **Preterm birth (<37 weeks of gestation)** | 1,071 (8.1) | 909 (9.8) | 12,485 (7.1) | 9,727 (7.8) |
| *Missing (%)* | *1.9* | *7.8* | *1.8* | *9.5* |
| *Prior health care utilisation/medical history* | | | | |
| **Number of GP contacts in the year before pregnancy** | | | | |
| 0 | 1,026 (7.6) | 1,731 (17.2) | 19,778 (11.0) | 31,981 (23.2) |
| 1-3 | 3,192 (23.7) | 2,235 (22.2) | 55,209 (30.7) | 38,835 (28.2) |
| 4-9 | 5,376 (39.9) | 3,400 (33.8) | 70,299 (39.1) | 44,061 (32.0) |
| ≥10 | 3,887 (28.8) | 2,686 (26.7) | 34,427 (19.2) | 22,779 (16.5) |
| **Median (IQR) number of GP contacts in the year before pregnancy** | 6.0 (3.0,10.0) | 5.0 (2.0,10.0) | 4.0 (2.0,8.0) | 3.0 (1.0,7.0) |
| **A&E visit for epilepsy at any point between the year before pregnancy and prior to the SWC or index date** | 265 (2.0) | 207 (2.1) | n/a | n/a |
| **Unplanned hospital admission for epilepsy at any point between the year before pregnancy and prior to the SWC or index date** | 1,007 (7.5) | 1,009 (10.0) | n/a | n/a |
| **Prescribed prophylactic contraception at any point between the year before pregnancy and prior to the SWC or index date** | 5,060 (37.5) | 3,904 (38.8) | 66,609 (37.1) | 51,756 (37.6) |
| **Prescribed emergency contraception at any point between the year before pregnancy and prior to the SWC or index date** | 497 (3.7) | 395 (3.9) | 5,364 (3.0) | 4,360 (3.2) |
| **Depression &/or anxiety diagnosis at any point between the year before pregnancy and prior to the SWC or index date** | 1,760 (13.1) | 1,231 (12.2) | 16,116 (9.0) | 11,526 (8.4) |
| **Urinary &/or faecal incontinence at any point between the year before pregnancy and prior to the SWC or index date** | 165 (1.2) | 91 (0.9) | 1,482 (0.8) | 967 (0.7) |
| **Dyspareunia, perineal &/or pelvic pain at any point between the year before pregnancy and prior to the SWC or index date** | 487 (3.6) | 205 (2.0) | 5,000 (2.8) | 2,463 (1.8) |

ᵃPercentage of those with complete data.

Abbreviations: GP, General practitioner; IMD, Index of Multiple Deprivation; IQR, interquartile range; SWC, postnatal six-week check.

of women being prescribed prophylactic and emergency contraception and having urinary and/or faecal incontinence or dyspareunia, perineal and/or pelvic pain recorded in the first year postpartum, with no evidence these associations differed according to whether or not a woman had epilepsy. Not having a SWC was also associated with a lower likelihood of women having depression and/or anxiety recorded in the first year postpartum among those without but not with epilepsy. Not having a SWC did not appear to affect the subsequent likelihood of women with epilepsy having epilepsy relevant adverse health outcomes observed in the first year postpartum after accounting for confounding by socio-demographic factors.

*(Continued)*

**Table 4. Among whole study population, general postnatal health outcomes in those who did not compared to did have a maternal SWC.**

| Outcomes at or after SWC or index date up to 1 year postpartum | Had a maternal SWC number of events/ person-years (rate per 100 person-years) | Did not have a maternal SWC number of events/ person-years (rate per 100 person-years) | Unad-justed HR (95 CI)¥ | Model A* HR (95% CI)¥ | Model B** HR (95% CI)¥ | Model C*** HR (95% CI)¥ | Model D**** HR (95% CI)¥ |
|---|---|---|---|---|---|---|---|
| Prescribed prophy-lactic contraception | 103,531/84,726 (122.20) | 55,388/88,423 (62.64) | **0.61 (0.61-0.62) p<0.001** | **0.61 (0.61-0.62) p<0.001** | **0.60 (0.60-0.61) p<0.001** | **0.60 (0.60-0.61) p<0.001** | **0.59 (0.58-0.60) p<0.001** |
| Prescribed emergency contraception | 5,569/159,350 (3.49) | 3,843/121,530 (3.16) | 0.96 (0.92–1.01) p=0.086 | 0.98 (0.94-1.03) p=0.425 | **0.92 (0.88-0.96) p<0.001** | **0.91 (0.87–0.95) p<0.001** | **0.95 (0.91–0.99) p=0.025** |
| Depression &/or anxiety | 20,540/152,030 (13.51) | 13,103/117,370 (11.16) | **0.85 (0.83-0.87) p<0.001** | **0.87 (0.85-0.89) p<0.001** | **0.84 (0.82-0.86) p<0.001** | **0.83 (0.81-0.85) p<0.001** | **0.88 (0.86-0.90) p<0.001** |
| Urinary &/or faecal incontinence | 1,944/162,100 (1.20) | 868/123,830 (0.70) | **0.57 (0.52–0.63) p<0.001** | **0.60 (0.55–0.66) p<0.001** | **0.63 (0.57–0.69) p<0.001** | **0.63 (0.58–0.69) p<0.001** | **0.67 (0.61–0.73) p<0.001** |
| Dyspareunia, perineal &/or pelvic pain | 2,896/161,820 (1.79) | 1,318/123,640 (1.07) | **0.60 (0.56–0.65) p<0.001** | **0.62 (0.57–0.66) p<0.001** | **0.63 (0.58–0.67) p<0.001** | **0.64 (0.59–0.68) p<0.001** | **0.70 (0.65-0.75) p<0.001** |

*Model A adjusted for year of birth only.

**Model B adjusted for year of birth and maternal socio-demographic characteristics (age at delivery, ethnic group, geographic region & IMD).

***Model C adjusted for variables in Model B and additionally adjusted for pregnancy/birth characteristics (parity, multifetal pregnancy, gestational hypertension or pre-eclampsia, mode of birth & preterm birth).

****Model D adjusted for variables in Model C and additionally adjusted for prior health care utilisation/medical history (number of GP contacts in the year before pregnancy and whether outcome in question was recorded at any point between the year before pregnancy and prior to the SWC or index date).

¥Analysis of all outcomes restricted to the population who did not have missing data for any of the covariates included in model D. Analysis of prophylactic contraception and emergency contraception additionally restricted to the women who did not have codes in their primary care records at or after the pregnancy end but prior to the SWC or index date indicating they could not become pregnant (outcomes prophylactic and emergency contraception n=292,606; all other outcomes n=293,272).

Abbreviations: CI, confidence interval; GP, General practitioner; HR, Hazard ratio; IMD, Index of Multiple Deprivation; SWC, postnatal six-week check.

## Strengths and limitations

Key strengths of this study includes its large population-based design, with the CPRD Aurum database considered to be broadly representative of the English population in terms of geographical spread, deprivation, ethnicity, age and gender [11,12]. The characteristics of the identified pregnancies in our study population were also broadly in line with the available national data on births in the UK [22–24]. We were also able to examine a range of outcomes. However, as with all studies using routinely-collected data, we are reliant on the quality of the data and the sensitivity/specificity of the code lists to identify the conditions and characteristics of interest. While we developed comprehensive codelists with clinical input and used validated codelists where available, we cannot rule out the possibility that some misclassification of our variables occurred, recognising that conditions/characteristics may not always be reported to or accurately recorded by healthcare professionals. This could have affected our SWC prevalence estimates and potentially biased our association estimates towards the null or under- or overestimated our association estimates depending on whether any misclassification in our exposures or outcomes was random or systematic in nature, respectively. Also, while we were able to adjust for a wide range of potential confounding factors including indicators of health-seeking behaviour and medical history, we cannot rule out the possibility of residual confounding due to mis-measured or unmeasured confounding factors such as educational levels and family support.

**Table 5. Among population with epilepsy, epilepsy relevant health outcomes in those who did not compared to did have a maternal SWC.**

| Outcomes at or after SWC or index date up to 1 year postpartum | Pregnancies to women with epilepsy who had a maternal SWC number of events/ person-years (rate per 100 person-years) | Pregnancies to women with epilepsy who did not have a maternal SWC number of events/ person-years (rate per 100 person-years) | Unadjusted HR (95 CI)¥ | Model A* HR (95% CI)¥ | Model B** HR (95% CI)¥ | Model C*** HR (95% CI)¥ | Model D**** HR (95% CI)¥ |
|---|---|---|---|---|---|---|---|
| A&E visit for epilepsy | 80/7,603 (1.05) | 63/5,002 (1.26) | 1.26 (0.89-1.79) p=0.200 | 1.25 (0.88-1.79) p=0.207 | 1.14 (0.80-1.64) p=0.467 | 1.13 (0.79-1.62) p=0.515 | 1.05 (0.73-1.52) p=0.781 |
| Unplanned hospital admission for epilepsy | 331/11,432 (2.90) | 359/8,482 (4.23) | **1.20 (1.01–1.42) p=0.035** | **1.23 (1.04–1.46) p=0.016** | 1.15 (0.96–1.36) p=0.123 | 1.13 (0.95–1.34) p=0.153 | 1.08 (0.91-1.28) p=0.379 |
| Mortality | 206/11,581 (1.78) | 246/8,645 (2.85) | 1.21 (0.96–1.52) p=0.111 | 1.16 (0.92–1.46) p=0.217 | 1.07 (0.85–1.35) p=0.572 | 1.07 (0.85–1.35) p=0.560 | 1.10 (0.87–1.39) p=0.427 |

*Model A adjusted for year of birth only.

**Model B adjusted for year of birth and maternal socio-demographic characteristics (age at delivery, ethnic group, geographic region & IMD).

***Model C adjusted for variables in Model B and additionally adjusted for pregnancy/birth characteristics (parity, multifetal pregnancy, gestational hypertension or pre-eclampsia, mode of birth & preterm birth).

****Model D adjusted for variables in Model C and additionally adjusted for prior health care utilisation/medical history (number of GP contacts in the year before pregnancy and whether outcome in question was recorded at any point between the year before pregnancy and prior to the SWC or index date).

¥Analysis of all outcomes restricted to the population who did not have missing data for any of the covariates included in model D. Analysis of A&E visit for epilepsy additionally restricted to pregnancies ending between 01/03/2007–31/03/2019 due to the availability of linked HES A&E data (outcome A&E visit for epilepsy n=13,325; all other outcomes n=20,531).

Abbreviations: A&E, Accident & Emergency; CI, confidence interval; GP, General practitioner; HES, Hospital Episode Statistics; HR, Hazard ratio; IMD, Index of Multiple Deprivation; SWC, postnatal six-week check.

## Comparison with other studies and interpretation

To the best of our knowledge, this is the first study to have examined the prevalence of the maternal SWC in women with epilepsy, how this compares to the prevalence seen in a general postnatal population sample of women without epilepsy, and whether this check is associated with subsequent maternal health outcomes in the first year postpartum. While it is reassuring that women with epilepsy appear to be as likely as those without epilepsy to have a SWC, this study suggests that a sizeable proportion of women (around 2 in 5) do not have a maternal SWC and there are other differences in the characteristics of those who do and do not have this check. This could reflect issues with the provision or uptake of the check. For example, some GP surgeries may not have routinely offered the check, at least until it became an essential service, women may find it difficult to access appointments or may not feel the need to have the check. A previous study [16] using the CPRD GOLD database also found some inequality in who had a SWC and reported a SWC prevalence estimate of 62% among women in England who gave birth between 2015 and 2018, which is comparable with the prevalence estimates seen in the equivalent time period in our study. Our prevalence estimates are also comparable to the SWC prevalence estimate of 56% reported among women in the UK who gave birth between 2006 and 2016 in a previous study [17] that used the Health Improvement Network (THIN) primary care database. However, the 2018 National Maternity Survey [25] found a higher prevalence, with 91% of women self-reporting a postnatal check-up of their own health with their GP. This discrepancy may be due to genuine checks not been recorded as such in the primary care records, although it is also possible the Maternity Survey overestimated the prevalence. This may have arisen if survey responders were more proactive in engaging with healthcare services and/or women incorrectly recalled any consultation with their GP in the postnatal period as a SWC. Indeed, previous studies [16,17] suggest around 80–90% of women have any consultation with their GP at the time of the postnatal check, which is closer to the National Maternity Survey estimate.

While we cannot rule out the possibility residual confounding could explain the observed differences in outcomes between women who did and did not have a SWC, another plausible explanation is this check may facilitate a conversation about contraception and other health needs. This could also empower women to seek contraception or medical help for health needs improving provision or detection. Indeed, a previous study [26] found that many health needs such as contraception, depression and low mood were documented in women's primary care records between 5–10 weeks postpartum corresponding to around the time many postnatal checks occur [16]. Our finding that not having a SWC is associated with a lower likelihood of having depression and/or anxiety recorded in the first year postpartum among those without but not with epilepsy might be due to increased monitoring and/or awareness of these mental health problems in women with epilepsy such that these issues tend to be picked up in these women regardless of whether or not they have a SWC. Recognising that women with epilepsy are at increased risk of depression in the postnatal period, UK guidelines [4] explicitly state that women with this condition should be screened for depressive disorder in the postpartum period and should be informed about the symptoms and provided with contact details for any assistance.

Of importance to highlight, regardless of whether or not women had a SWC, the rates of general postpartum adverse health outcomes observed in the primary care records, at least for urinary and/or faecal urinary and/or faecal incontinence and dyspareunia and perineal and/or pelvic pain, appear to be on the low side given the prevalence of postpartum health problems that have been reported in the literature [25,27]. This could reflect the fact that previous studies have largely relied on surveys where postpartum women are asked directly about and self-report conditions or symptoms. This could pick up issues that women do not report to their GP for various reasons such as they are reluctant to, do not feel the need to, do not have the time to do, or have sought help for elsewhere. The guidance on administrating the SWC does list physical and pelvic issues among the topics to discuss, however we cannot tell if GPs are proactively asking women or if they assume women will report any significant problems. Women who do not have a SWC are perhaps even less likely to report problems like incontinence or dyspareunia, since it requires them to make an appointment to do so. We also cannot rule out we may have underestimated outcomes due to coding issues in the GP records. While comprehensive codelists were used to identify the outcomes, informed by clinical input and searching published articles and online codelist repositories, it's possible the codes used had low sensitivity for identifying the outcomes although validation studies are needed to verify this.

## Conclusions and implications

This study found that women with epilepsy were as likely as the general postnatal population sample of women without epilepsy to have evidence of a maternal SWC. However, around 2 in every 5 women had no evidence of a maternal SWC and there were other differences in the characteristics of those who did and did not have this check which could reflect issues with provision or uptake. In 2020, the maternal SWC became an essential service in England under the GP contract [2] and clearer guidance is now available [28] on the conduct and content of this check which should hopefully lead to improvements. We also found that regardless of whether or not a women had epilepsy, women who did not have a maternal SWC were less likely than those who did have this check to subsequently be prescribed prophylactic and emergency contraception and have various general adverse postpartum health outcomes recorded in their primary care records in the first year postpartum. These findings suggest the maternal SWC may play a role in increasing the use of contraception and the detection or treatment of adverse health outcomes in the first year postpartum.

## Supporting information

**S1 Table. Derivation of data items of interest.**
(DOCX)

**S1 Fig. Flow diagram of identification of study population.**
(PDF)

**S2 Table. Characteristics of complete case and whole study population.**
(DOCX)

**S3 Table. General postpartum health outcomes in those who did not compared to did have a maternal SWC by epilepsy status.**
(DOCX)

## Acknowledgments

We would like to thank all those involved in recruiting and participating in our patient and public involvement, including Alison Cooper-Goodison and Epilepsy Action. We also wish to thank Dr Sarah Hillman for her help in developing the code list to identify of urinary and/or faecal incontinence in the primary care data.

## Author contributions

**Conceptualization:** Kathryn E. Fitzpatrick, Claire Carson.

**Data curation:** Chun Hei Kwok.

**Formal analysis:** Kathryn E. Fitzpatrick.

**Funding acquisition:** Fiona Alderdice, Chris Gale, Sara Kenyon, Maria A Quigley, Julia Sanders, Dimitrios Siassakos, Claire Carson.

**Methodology:** Kathryn E. Fitzpatrick, Liza Bowen, Yangmei Li, Fiona Alderdice, Suresha Dealmeida, Chris Gale, Sara Kenyon, Maria A Quigley, Julia Sanders, Dimitrios Siassakos, Claire Carson.

**Project administration:** Kathryn E. Fitzpatrick.

**Supervision:** Claire Carson.

**Writing – original draft:** Kathryn E. Fitzpatrick.

**Writing – review & editing:** Liza Bowen, Yangmei Li, Chun Hei Kwok, Fiona Alderdice, Suresha Dealmeida, Chris Gale, Sara Kenyon, Maria A Quigley, Julia Sanders, Dimitrios Siassakos, Claire Carson.

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
