## [Decision Letter · Decision Letter 0]

3 Mar 2025

PONE-D-24-44685The maternal postnatal six-week check in women with epilepsy: does the prevalence or subsequent postpartum health differ from the general postnatal population?PLOS ONE

Dear Dr. Fitzpatrick,

Thank you for submitting your manuscript to PLOS ONE. After careful consideration, we feel that it has merit but does not fully meet PLOS ONE’s publication criteria as it currently stands. Therefore, we invite you to submit a revised version of the manuscript that addresses the points raised during the review process.

**Please note that we have only been able to secure a single reviewer to assess your manuscript. We are issuing a decision on your manuscript at this point to prevent further delays in the evaluation of your manuscript. Please be aware that the editor who handles your revised manuscript might find it necessary to invite additional reviewers to assess this work once the revised manuscript is submitted. However, we will aim to proceed on the basis of this single review if possible. **

**Could you please revise the manuscript to carefully address the concerns raised?**

We look forward to receiving your revised manuscript.

Kind regards,

Helen Howard

Staff Editor

PLOS ONE

**Journal Requirements:**

Please ensure that your manuscript meets PLOS ONE's style requirements, including those for file naming. The PLOS ONE style templates can be found at https://journals.plos.org/plosone/s/file?id=wjVg/PLOSOne_formatting_sample_main_body.pdf and https://journals.plos.org/plosone/s/file?id=ba62/PLOSOne_formatting_sample_title_authors_affiliations.pdf 2. Please note that PLOS ONE has specific guidelines on code sharing for submissions in which author-generated code underpins the findings in the manuscript. In these cases, we expect all author-generated code to be made available without restrictions upon publication of the work. Please review our guidelines at https://journals.plos.org/plosone/s/materials-and-software-sharing#loc-sharing-code and ensure that your code is shared in a way that follows best practice and facilitates reproducibility and reuse. 3. Thank you for stating the following financial disclosure:  This research is funded by the National Institute for Health Research (NIHR) Policy Research Programme, conducted through the NIHR Policy Research Unit in Maternal and Neonatal Health and Care, PR-PRU-1217-21202. SK is part funded by the NIHR Applied Research Collaboration in the West Midlands. The views expressed are those of the authors and not necessarily those of the NIHR or the Department of Health and Social Care.  Please state what role the funders took in the study.  If the funders had no role, please state: "The funders had no role in study design, data collection and analysis, decision to publish, or preparation of the manuscript." If this statement is not correct you must amend it as needed. Please include this amended Role of Funder statement in your cover letter; we will change the online submission form on your behalf. 4. We note that you have indicated that there are restrictions to data sharing for this study. For studies involving human research participant data or other sensitive data, we encourage authors to share de-identified or anonymized data. However, when data cannot be publicly shared for ethical reasons, we allow authors to make their data sets available upon request. For information on unacceptable data access restrictions, please see http://journals.plos.org/plosone/s/data-availability#loc-unacceptable-data-access-restrictions.  Before we proceed with your manuscript, please address the following prompts: a) If there are ethical or legal restrictions on sharing a de-identified data set, please explain them in detail (e.g., data contain potentially identifying or sensitive patient information, data are owned by a third-party organization, etc.) and who has imposed them (e.g., a Research Ethics Committee or Institutional Review Board, etc.). Please also provide contact information for a data access committee, ethics committee, or other institutional body to which data requests may be sent. b) If there are no restrictions, please upload the minimal anonymized data set necessary to replicate your study findings to a stable, public repository and provide us with the relevant URLs, DOIs, or accession numbers. Please see http://www.bmj.com/content/340/bmj.c181.long for guidelines on how to de-identify and prepare clinical data for publication. For a list of recommended repositories, please see https://journals.plos.org/plosone/s/recommended-repositories. You also have the option of uploading the data as Supporting Information files, but we would recommend depositing data directly to a data repository if possible. Please update your Data Availability statement in the submission form accordingly.

Reviewers' comments:

Reviewer's Responses to Questions

**Comments to the Author**

1. Is the manuscript technically sound, and do the data support the conclusions?

Reviewer #1: Yes

2. Has the statistical analysis been performed appropriately and rigorously? 

Reviewer #1: Yes

3. Have the authors made all data underlying the findings in their manuscript fully available?

Reviewer #1: Yes

4. Is the manuscript presented in an intelligible fashion and written in standard English?

Reviewer #1: Yes

5. Review Comments to the Author

**Reviewer #1:**  These researchers sought to determine if there is a difference in attendance at a six-week postpartum check among patients with and without epilespsy. Further if patients attended a SWC, were there differences in mental health outcomes or epilepsy-related complications. They used 4 administrative databases/registries to evaluate this question. Specific codelists were previously used to identify this population and healthcare visits. The statistical analysis seems appropriate using Poisson regression modeling as the outcome of a SWC if very common. Several models were fit to evaluate for possible important associations. Importantly, appropriate confounders were selected based on either prior evidence or hypotheses for confounding effects. Patients identified as having epilepsy were relatively similar to those without, apart from number of prior health care visits prior to pregnancy, and index of multiple deprivation.

Overall, the researchers used rigorous methodology to answer challenging but meaningful questions.

SPECIFIC POINTS:

INTROUDCTION:

Line 128 - Describe term A&E

METHODS:

Line 150 - Describe "Stillbirth" and how this is captured differently than miscarriage/clinical pregnancy loss

RESULTS:

Line 265-266 - the median age between the two groups is nearly identical. Similarly, prior prescription of prophylactic or emergency contraception, urinary/faecal incontinence and dyspareunia/pelvic pain are very similar between the groups. I assume these differences stated are based on a statistically significant p-value (which is not reported), and largely because of the large sample size rather than clinically meaningful differences between the groups.

DISCUSSION:

Line 511 - I think you need to consider more strongly that GPs may not be assessing symptoms of dyspareunia and other symptoms of pelvic floor dysfunction adequately. This may be a reflection of a patient's discomfort with their GP, which is why they are not attending the SWC and therefore have a lower reported incidence. Alternatively, it is a poorly coded variable. Have there been any validation studies of the codes you are using to identify these symptoms with accuracy? I suspect these results are largely underrepresented in the way these data are collected.

6. PLOS authors have the option to publish the peer review history of their article (what does this mean? ). If published, this will include your full peer review and any attached files.

**Do you want your identity to be public for this peer review?** For information about this choice, including consent withdrawal, please see our Privacy Policy .

Reviewer #1: No

---

## [Author Response · Author response to Decision Letter 1]

25 Mar 2025

Please see uploaded file 'Response to Reviewers'.

---

## [Editor Report · Decision Letter 1]

2 Apr 2025

The maternal postnatal six-week check in women with epilepsy: does the prevalence or subsequent postpartum health differ from the general postnatal population?

PONE-D-24-44685R1

Dear Dr. Fitzpatrick,

We’re pleased to inform you that your manuscript has been judged scientifically suitable for publication and will be formally accepted for publication once it meets all outstanding technical requirements.

Kind regards,

Chun Liu

Academic Editor

PLOS ONE
---

## [Editor Report · Acceptance letter]

PONE-D-24-44685R1

PLOS ONE

Dear Dr. Fitzpatrick,

I'm pleased to inform you that your manuscript has been deemed suitable for publication in PLOS ONE. Congratulations! Your manuscript is now being handed over to our production team.

Kind regards,

on behalf of

Dr. PLOS Manuscript Reassignment

Staff Editor

PLOS ONE